# Fake News in the Post-COVID-19 Era? The Health Disinformation Agenda in Spain

Carmen Costa-Sánchez [1,*], Ángel Vizoso [2] and Xosé López-García [2]

1 Grupo de Investigación Cultura e Comunicación Interactiva, Facultade de Ciencias da Comunicación, Universidade da Coruña, Campus de Elviña, 15701 A Coruña, Spain

2 Grupo de Investigación Novos Medios, Facultade de Ciencias da Comunicación, Universidade de Santiago de Compostela, Avenida de Castelao, 15782 Santiago de Compostela, Spain; angel.vizoso@usc.es (Á.V.); xose.lopez.garcia@usc.es (X.L.-G.)

* Correspondence: carmen.costa@udc.es

**Abstract:** Three years after a pandemic that demonstrated the importance of reliable health information in a news agenda dominated by coronavirus disease 2019 (COVID-19), we analyze the situation of health disinformation in Spain on the basis of the verifications carried out by its main fact-checking platforms. The results show that COVID-19 shared center stage with other topics in the health area. In addition, a unique agenda is evident in each situation in the study, indicating a fact-checking strategy that is differentiated according to the media outlet and type of specialization (generalist fact-checker or one specialized in health). Vaccination, nutrition, and disease treatment emerge as the most important thematic subfields. Most health hoaxes are manufactured, i.e., created from scratch, rather than being manipulated or reconfigured from real preexisting elements. The format of text and image together predominates, and new social networks (TikTok or Telegram) have appeared as platforms for the circulation of hoaxes. This indicates that providing necessary health literacy to society and giving health issues greater presence in current fact-checking agendas are strategies for combatting disinformation, which can have serious consequences, regardless of whether there is a public health crisis such as the one experienced recently.

**Keywords:** fact checking; COVID-19; health; crisis; agenda; disinformation; Spain



## 1. Introduction

The coronavirus disease 2019 (COVID-19) pandemic revealed the engineering of disinformation on a global level, with confusing and unfounded messages circulating through different communication channels, messaging platforms, and social networks; this had tremendously adverse effects at different levels [1]. Society worldwide experienced moments of enormous uncertainty, exactly the context in which truthful and reliable information is more necessary than ever [2]. Journalism addressed this unprecedented crisis by increasing its informative efforts in this area, striving to identify the main social concerns and administering "vaccines" in the form of both news coverage and fact checking [3,4].

"Closed" social and instant messaging networks (such as WhatsApp) have been identified as the main platforms through which fake news was circulated during the pandemic [5]. The consequences of this disinformation phenomenon are not minor. Previous studies have shown that the proliferation of fake news has consequences for public health because it fuels panic among people and discredits the scientific community in the eyes of the public [6]. As a result of what happened, some authors have indicated that it is necessary for journalism to reinforce its commitment to trustworthy and fact-checked news, in collaboration with other social actors such as fact-checking platforms [7].

Now, 3 years after the World Health Organization (WHO) declared an infodemic [8], COVID-19 has ceased to make headlines and has stepped out of the media spotlight. As stated in the 2021 Quiral Report [9], in the period from January 2021 to May 2022, COVID-19

has continued to be present in the media but has now taken a back seat, in terms of both the quantity of news items and their position in the news hierarchy, with a gradual decrease in interest punctuated by occasional moments of news impact related to specific news items. Terms such as "influenzaization" and "post-pandemic" have begun to become more prominent, conveying to the public the idea of a new phase after the pandemic. However, new cases continue to be detected, there is still no answer for persistent COVID-19, and the situation varies between countries with different healthcare settings. In this ever-changing situation in which COVID-19 is still not behind us and its consequences persist, it is necessary to determine how the infodemic is evolving [8] and understand its coexistence with other forms of disinformation in health news and journalism.

### 1.1. COVID-19 and Disinformation

As shown before, misinformation is a challenge for both society in general and the media. In the years leading up to the pandemic, it was possible to witness both the increase in false information circulating mainly through spaces such as social networks and the emergence of numerous initiatives specializing in its verification [10]. However, if we look closely at the period that began with the mass confinement of the population in the first months of February 2020, the target of disinformation (and also of information) is focused on one content above many others: COVID-19 [11].

Within the pandemic itself, the high media exposure resulting from the lockdown of the population and the exceptional measures taken by governments around the world led to a large increase in the circulation of false content. Moreover, the misinformation about the measures taken by governments, the figures provided, the impact on the economy, etc., led to a growing mistrust of the official version [12]. This can be observed in the global context, but also in the particular contexts of each country [13].

In the central months of the pandemic, these hoaxes took all sorts of forms and were spread through multiple channels, although they found their main home in social networks and instant messaging applications [12]. Within the latter, WhatsApp was the main one [14]. In this application, fakes took advantage of capabilities such as sending audio or images to produce a greater amount of completely false content, while other types of hoaxes were created on media such as social networks.

At the same time, the volume of verifications of this type of disinformation also increased, adapting narratives and using new channels for the publication of truthful elements [15]. This circumstance of adaptation and the effort to reach the audience is not minor, as it is one of the key elements in trying to ensure the success of a fact-check [16]. Therefore, during the central months of the outbreak, it was possible to observe the use of channels and languages already used until then for disinformation, although with some transformations in terms of their use.

### 1.2. Disinformation about Health in Spain

Health disinformation existed before the Internet and social networks. In Spain, historically, there was a well-known incident surrounding the spread of cholera in 1834, during which the facts were spiced up with fake news. It was suggested that clergy were contaminating water to cause this disease that was so little understood and spread so quickly and lethally, triggering a massacre of friars; specifically, about 60 clergymen were killed on 17 July 1834. Other historical examples confirm the intentional use of the disinformation–health binomial for various types of manipulative purposes [17].

Observed in a broader context, there is evidence of the use of disinformation as far back as the Roman Empire [18]. Nonetheless, advances in communication such as the invention of the printing press and the increasing ease of disseminating content led to a wider spread of hoaxes. However, it is in recent times that this misinformation has taken on greater speed and importance in multiple conditions. The emergence of Artificial Intelligence has had a notable impact not only on the production and dissemination of hoaxes [19], but also on its identification and verification [20]. The clearest examples of

this are deepfakes, highly realistic audio or video pieces created through manipulation using this type of AI-based tools [21]. All of this poses a challenge for the media and large Internet platforms [22], as it makes it necessary to establish protocols for the verification and denial of these false contents.

Nowadays, in a scenario driven by the Internet and platforms, health disinformation remains a factor that not only exists but is also on the rise [23].

In particular, research has been conducted on disinformative tactics used by climate change deniers [24] and anti-vaccine movements [25]. Campaigns in favor of alternative medicine and homeopathy, hoaxes related to supposedly miraculous weight loss diets, unsubstantiated messages about the effects of genetically modified foods, and the like have also been identified, although none of these issues has equaled the magnitude and impact of the disinformation related to the COVID-19 pandemic [5].

Since then, research has focused on what has happened during these 3 years regarding the pandemic and related and recurring issues, covering the virus's origin, the effects of vaccines, political actors and their statements, and institutional messages and actors [5,13,26–28].

However, as news related to the pandemic and the fact-checks carried out by fact-checkers has waned [29], there has been a lack of research that puts into perspective how the leading role occupied by the new virus is leaving room for other issues related to health—or in other words, how the agenda of "hoaxes versus fact checking" in health matters in Spain has evolved in the 3 years since the declaration of the global pandemic.

On the basis of the confirmed premise that the health agenda has been dominated by COVID-19 in multiple ways, it is necessary to understand its evolution, as the Spanish Council of Ministers has now declared the end of the public health crisis and eliminated the mandatory use of masks, while vaccination has taken a back seat.

Given the active listening channels that fact-checkers maintained as platforms specialized in fact checking misleading information through social networks and direct communication channels (via WhatsApp, in a form available to users, etc.), this leads to a reflection on the main health concerns of the Spanish public today.

Newtral and Maldita are the two main examples of independent fact-checking platforms operating in Spain [30]. Both were launched in 2018, being connected to the activities of experienced journalists. They are two of the four Spanish media signatories of the International Fact-checking Network, and the only two that are independent. #SaludsinBulos is the only example in Spain of a news fact-checking platform specialized in health issues. The main research question (and sub-questions) of this paper are, therefore, exploratory in nature:

What has happened with health disinformation since the decline of COVID-19 news? (R.Q.1.)

- How have COVID-19-related hoaxes evolved in the 3 years since the declaration of the infodemic? (R.Q.1.1.)
- What are the current topics on the main Spanish fact-checkers' health fact-checking agendas? (R.Q.1.2.)

## 2. Materials and Methods

The current analysis is based on the application of quantitative methods to elucidate the development of health-related hoaxes and their fact checking. This research continues the line of study opened by the authors in a previous paper in which they analyzed fake news and fact-checking processes in Spain during the COVID-19 pandemic [1].

This study seeks to identify the nature of the hoaxes disseminated in relation to the field of health once the restrictions on the COVID-19 pandemic in Spain have been lifted.

The starting premise for this study is that, once the restrictions on this disease have been lifted, the decline in information and public interest in it has led to its decline as the main topic of health disinformation. A second hypothesis would consequently point to the reappearance of other health-related misinformation that had hitherto remained in the background.

For this purpose, three fact-checkers in a recent time frame were selected. First, a review was carried out of all the hoaxes debunked by Newtral and Maldita between 15 January and 15 March 2023, extracting those related to health issues. These media outlets were selected in order to ensure the continuity of the study carried out at the height of the pandemic, and these two initiatives were chosen due to their relevance in the Spanish fact-checking landscape. With regard to the temporal selection, a sample of two months was chosen. The reason for this selection and also for the interval is to leave out holiday periods such as Christmas or Easter, which can bring with them other types of misinformation also in the field of health, but particular to those times. In addition, these two months offer the possibility of analyzing several weeks of fact-checkers' work, thus being able to observe the interest of these media in health-related hoaxes.

Furthermore, the same was performed for the fact-checker specialized in this subject, #SaludsinBulos, which combines news and advice on health and good habits with debunking of myths and hoaxes. Given that this space has a lower frequency of publication—but taking into account its great relevance to the topic being analyzed—a different period of analysis was chosen by reviewing all hoaxes from when their activity began (7 November 2017) until the last one identified (published on 22 February 2023).

The proposed analysis identifies and reviews 98 different hoaxes that were debunked and explained by the three fact-checkers studied: Newtral (*N* = 16), Maldita (*N* = 34), and #SaludsinBulos (*N* = 48). To understand their thematic characteristics and scope of impact, an analysis sheet designed by the authors was applied to each of them, including the following information:

1. Identifying the data of the fake news item (headline, media outlet that debunked it, date of publication, and URL).
2. Information regarding the content of the hoax and its fact checking:
   - Topic: After an initial exploration and after consulting previous studies [31–33], 14 possible areas into which health-related hoaxes fall were identified: (1) nutrition, diet, and food; (2) exercise, physical activity, and fitness; (3) psychology; (4) stress or anxiety; (5) cancer; (6) mental health; (7) sleep disorders; (8) disease treatment; (9) diagnosis and diagnostic testing; (10) sexual and reproductive health; (11) drugs; (12) COVID-19; (13) research funding and fundraising; and (14) government measures and health protocols.
   - COVID-19-specific disinformation: For those cases in which a hoax related to this disease was identified, a series of nine possible topics were established: (1) politicians; (2) measures and sanctions; (3) remedies, drugs, and vaccines; (4) hospitals and healthcare; (5) incidents; (6) scams and phishing; (7) grants, donations, and fundraising campaigns; and (8) political or commercial theories.
   - Sources used to fact-check the hoax: Within these, it is possible to differentiate scientific (scientific associations, health professionals, experts, medical societies), referential (WHO, alert coordination centers), and institutional (other non-health institutions) sources.
   - The thematic scope of disinformation, distinguishing between local, regional, national (Spain), and international.
   - The type of fake news: Reconfiguration—content left out, false context, or manipulated content; manufactured—fabrication or faked; parody; and deepening—those examples whose objective is to clarify some false beliefs that exist in society.
   - The format of the disinformation: image, text, video, infographic, or other.
   - The identifiable motivation behind the hoax: (1) journalistic error, (2) parody or satire, (3) troll, (4) political motives, (5) economic gain, or (6) unclear.
   - The main platform(s) through which this disinformation was reported to be distributed: Facebook, Twitter, YouTube, TikTok, Telegram, WhatsApp, Reddit, websites, or others.

In short, the analysis carried out provides insight into the development of disinformation in the health field now that the COVID-19 pandemic has been overcome and a large part of the specific measures for dealing with it have been withdrawn. The "Results" section describes the main characteristics of the spread of these hoaxes for the periods and the specialized media outlets analyzed.

## 3. Results

### 3.1. Quantitative Impact of COVID-19-Related Hoaxes

The temporal and thematic evolution (Figure 1) indicates an evident decrease in the impact of COVID-19-related hoaxes from the acute stage of the crisis (March 2020) to the later stages (August 2020 and the 2023 period). It should be noted that the third period of analysis, corresponding to 2023, is much longer than the other two periods analyzed (two months versus fortnights), so a greater quantitative impact could be expected. This is true for Maldita, which accumulated more COVID-19 debunkings than Newtral, an agent for which pandemic hoaxes accounted for 6% of its current health fact-checking agenda (versus 53% for Maldita).

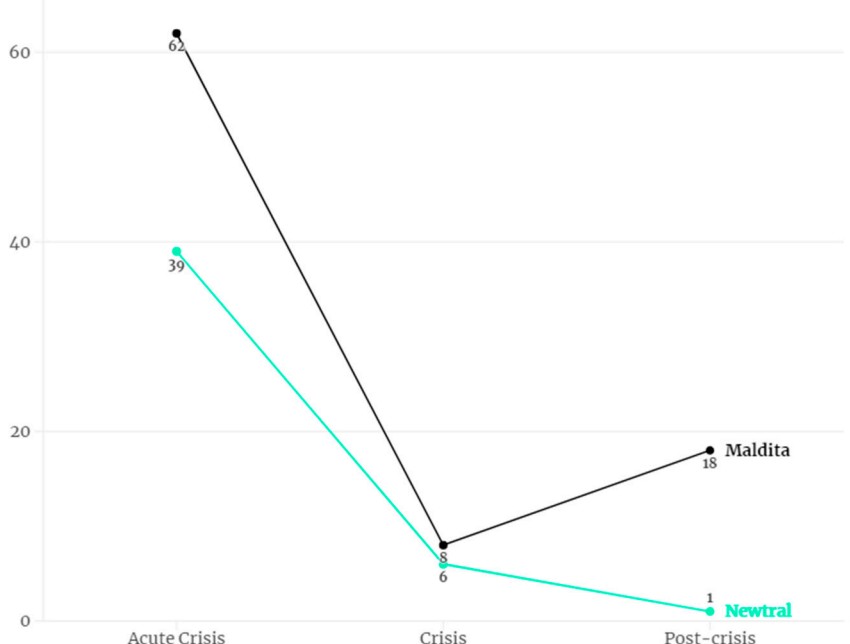

**Figure 1.** COVID-19 numeric impact. Authors' own elaboration.

We must therefore discuss the transition from a fact-checking agenda dominated by the pandemic in the initial and acute stage of the global public health crisis to a stabilization stage, in which there was a significant decrease for both platforms, to a third stage, in which each fact-checking agent creates a health fact-checking agenda with a different impact granted to COVID-19 in comparison with other health-related topics, with which the pandemic now shares center stage.

### 3.2. Themes around Health and Fact-Checking Agendas

The results reveal the construction of three differentiated agendas of health-related issues and their importance (Figure 2).

In the case of Maldita, COVID-19 continues to take up the lion's share of the health topics addressed in 2023. The subtopic with the most fact-checks in this regard is vaccines, which owing to the anti-vaccine movement continues to create false news that generates fear and suspicion around vaccination. Next in importance are issues related to sexual/reproductive health (18%) linked to current political events in Spain (either linked to the image or statements of political figures, or to a subtopic of the political agenda on chil-

dren, sexual health, and textbooks). In third place, in order of quantitative importance, are the topics of nutrition (food), also with a second political (meat consumption and Agenda 2030) or xenophobic (food from certain allegedly contaminated countries) interpretation, and hoaxes related to the treatment of diseases (with a clear educational dimension).

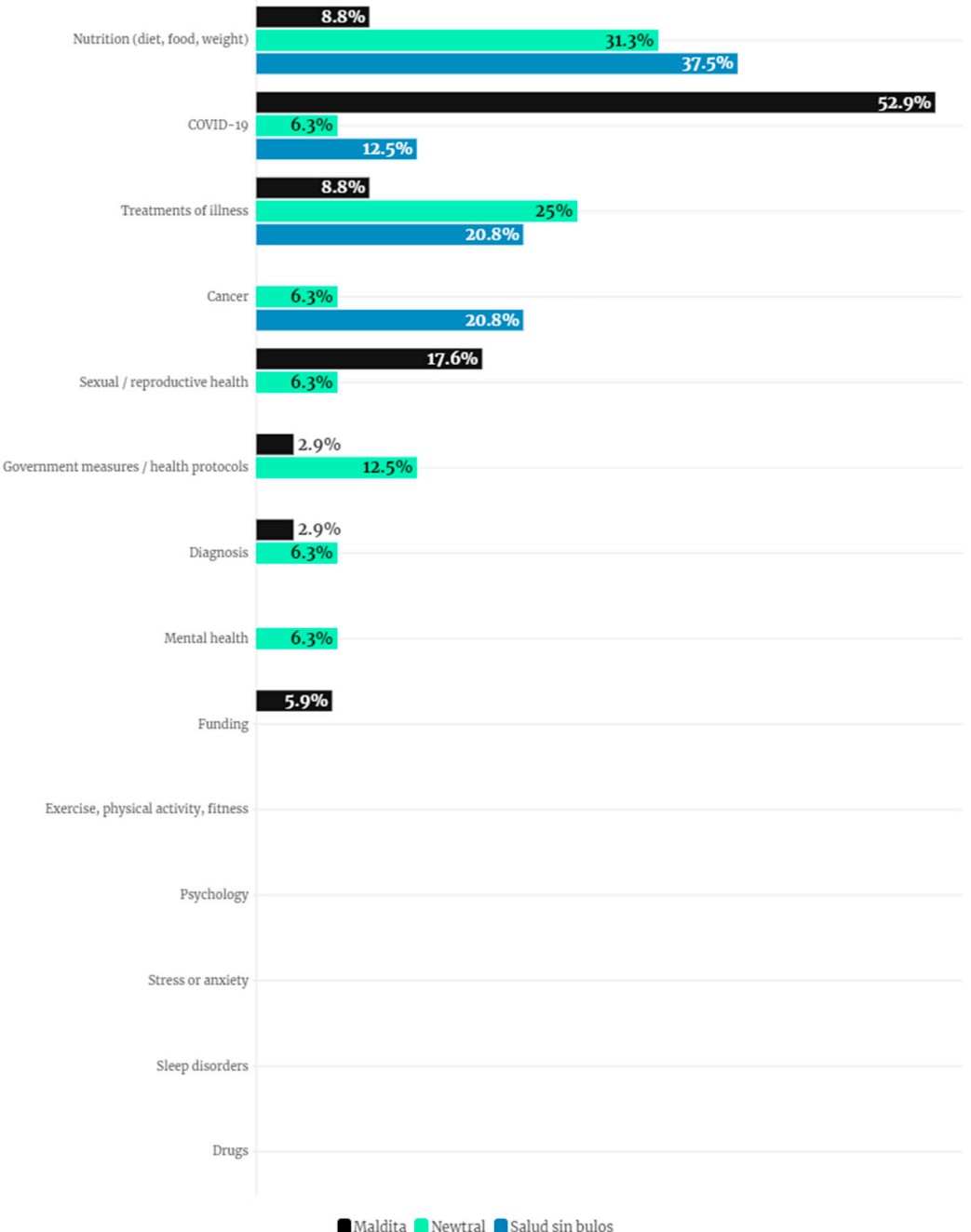

**Figure 2.** Health agenda issues on different fact-checking platforms. Authors' own elaboration.

In the case of Newtral, health-related fact-checks play a less significant role than in the case of Maldita. In this platform's fact-checking agenda, health is therefore of less importance in this 2023 analysis. Within it, nutrition was the main theme (from both an educational and health policy perspective) followed by government measures/health protocols.

In the analysis of the only specialized health fact-checking platform, viz. #SaludsinBulos, nutrition also ranked first, followed by cancer and treatment of diseases. COVID-19 is now in fourth position when it comes to topics. Specifically analyzing the subtopics

included in nutrition, the news dimension, especially in relation to the benefits of certain foods, is predominant, in addition to the subtopics of cancer and disease treatment. Among those related to COVID-19, vaccines are the main topic of the hoaxes analyzed.

### 3.3. Reliable Sources of Information

In this regard, in the previous study [1], in relation to the pandemic, we detected that the news leadership of institutional sources is fact-checking hoaxes, but with an upward push from scientific sources (experts, health professionals, and scientific associations). In the analysis of the health hoax agenda, the use of non-health institutional sources currently predominates (46.1%), followed at some distance by scientific sources (25%).

The analysis changes in the case of the specialized fact-checking platform, #Saludsin-Bulos, as this pattern is reversed, with scientific sources accounting for 48% of the sources used in fact-checking news, while non-health institutional sources accounted for only 4.3%.

### 3.4. Scope of Circulation

While the scope of circulation of disinformation during the pandemic was both national and international in equal measure, in the current health agenda, an international scope of circulation predominates (60%), with twice as many hoaxes as those impacting at the Spanish level (30%).

Combining the analysis of the scope of circulation and topic (Figure 3), it can be seen that, at the international level, COVID-19 continues to be the main topic, followed by hoaxes related to nutrition (food, diets, etc.). With respect to those with a national impact in Spain, the analysis showed that the breakdown of topics was more spread out, making sexual/reproductive health issues the most prevalent in relation to the political agenda.

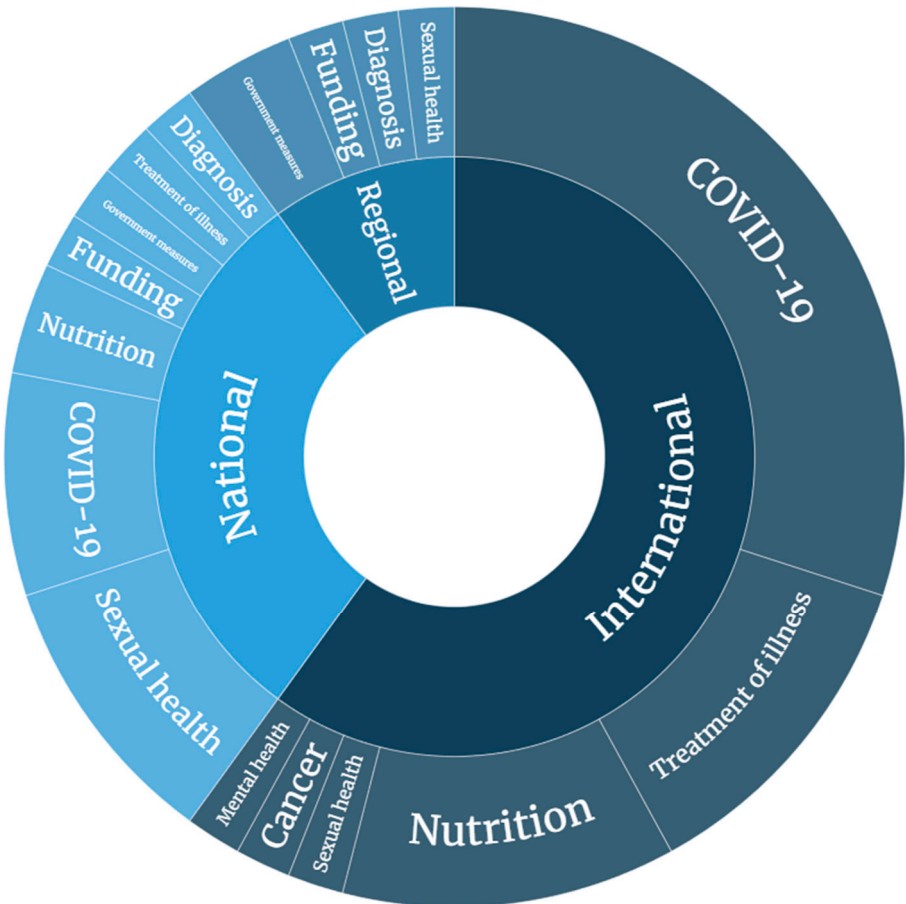

**Figure 3.** Disinformation by themes. Authors' own elaboration.

In the case of the specialized platform specifically, we again found a different pattern: the majority of disinformation was not limited to a specific geographical reference area, followed by those linked to the national level.

This occurs because the fact-checks linked to health information (without political or institutional dimensions, for example), which is the main focus of #SaludsinBulos, are difficult to limit to a particular scope of dissemination and geographic impact.

### 3.5. Types of Hoaxes and Formats

Of the four main types of hoax classification (reconfigured, manufactured, parody, or deepening), no cases of parody or deepening were found.

The results (Table 1) showed that health disinformation is mostly manufactured (74%), rather than being reconfigured or modified (22%), with rates similar to those detected for the COVID-19 subtopic. In this case, the trend is common to both types of fact-checkers analyzed (generalists and those specialized in health).

**Table 1.** Typology of hoaxes (global analysis).

| Type | Percentage (T) |
|---|---|
| Manufactured | 73.4% |
|     Fabrication | 79.1% |
|     Faked | 19.4% |
| Reconfigured | 26.5% |
|     Content left out | 30.7% |
|     Fake content | 53.8% |
|     Manipulated content | 15.3% |

Notes: Percentage (T) indicates the percentage out of 100% of the hoaxes analyzed by the three fact-checking agents. Source: Authors' own creation

Of the manufactured hoaxes, 79% are "pure" fabrication, whereas 20% would fall into the "faked" category. Of the reconfigured hoaxes, those that left out content accounted for 30% of the total number of hoaxes analyzed, while those with false content accounted for 53%.

In terms of format (Table 2), images were the predominant format for the health-related hoaxes fact-checked by all the platforms (46%), followed by text and image together (36%) or text alone. After that came videos, while no cases of disinformation in infographic format were detected.

**Table 2.** Format of hoaxes (global analysis).

| Format | Percentage (T) |
|---|---|
| Image | 46.1% |
| Video | 25% |
| Infographic | 0% |
| Text | 26.9% |
| Text + image | 36.5% |

Notes: Percentage (T) is the percentage of 100% of the hoaxes analyzed by the three fact-checking agents. Source: Authors' own creation.

### 3.6. Platforms

Regarding the platforms through which the hoaxes were circulated (Figure 4), in the case of the specialized health fact-checker, WhatsApp appears to be the main platform through which disinformation circulated.

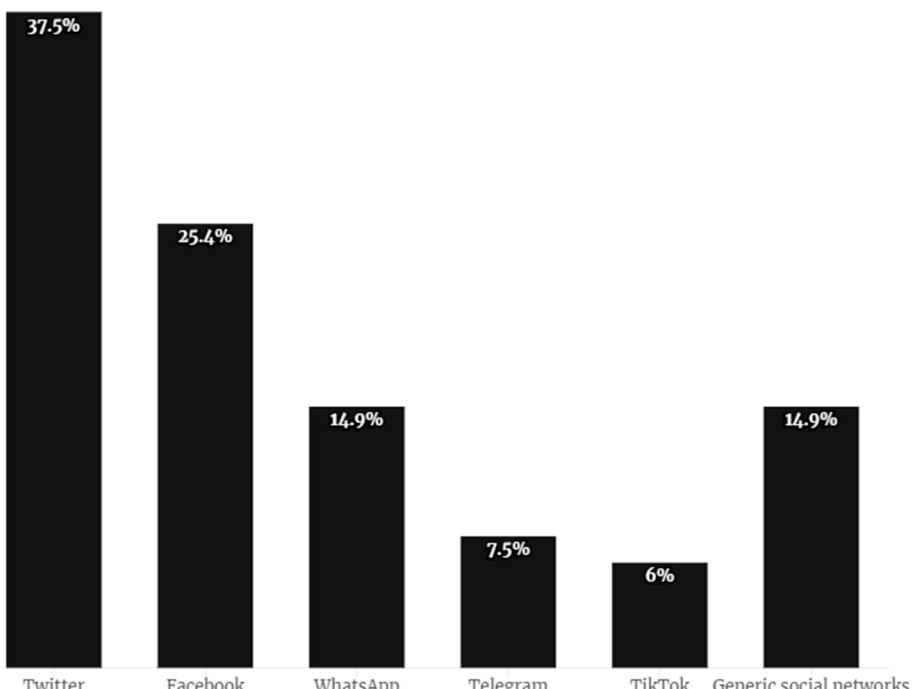

**Figure 4.** Platforms where hoaxes were debunked. Authors' own elaboration.

In the case of generalist fact-checkers (Newtral and Maldita), Twitter, Facebook, and WhatsApp were identified as the main platforms through which debunked health hoaxes circulated.

Also note the rise of new networks such as Telegram or TikTok, which the Alpha generation in Spain in particular use, and which appear in this analysis as emerging channels for the spread of health disinformation. Another portion of the fact-checks did not indicate a specific network or platform but rather a generic designation (social networks).

*3.7. Interest/Drivers*

The hidden motivations or elements that may be driving health disinformation are difficult to detect, and they can also work in combination. However, in the generalist media, political motivations and interests (72%) are clearly and mainly evident, followed by trolling (22%) and economic benefit (4%).

**4. Discussion and Conclusions**

The present study is not just a continuation of previous studies, focused on the cataloging and evolution of COVID-19 hoaxes in Spain, more than three years after the global pandemic. Although this is one of the aspects discussed, this research aims to place health within the framework of misinformation in Spain, which would encompass those hoaxes related to COVID-19, but is not limited to them. This represents a uniqueness that differentiates this work from previous research exclusively focused to date on the relationship between disinformation and pandemics. There is limited recent scientific literature on misinformation and health that specifically analyzes the Spanish area [34,35], a research gap that this work contributes to addressing. Obviously, in recent years, the severity and novelty of the pandemic directed the media focus, citizen concern, and health misinformation to a single topic agenda: COVID-19. Readjusted to post-COVID-19 normality, the research shows that the themes of health misinformation change and that they are readjusted according to the editorial strategies of each media.

The results of this study indicate, first, that health disinformation occupies a small place in the fact-checking agenda of the main platforms in Spain (with an average of 6 fact-checked items of disinformation per month in 2023). At present, each fact-checking

platform also applies its own health fact-checking agenda, with more or less of a significant commitment to this issue. In the period analyzed, for example, Maldita was more committed to combating this type of disinformation than Newtral. It should also be noted that Maldita has a section specialized in nutrition (Maldita Nutrition), although it has not been included in this analysis so as not to generate a clear differentiating bias with respect to the results of this platform, thus altering any comparative methodology. In any case, it is evident that making a firm commitment to this issue of huge social relevance depends on the editorial policy of each fact-checking media outlet.

This individualized agenda also applies to disinformation about COVID-19. In this sense, first of all, it is worth concluding that in this post-pandemic period, the number of hoaxes about COVID-19 has decreased comparatively with previous studies [1,5,12,36,37], which has given rise to new topics in the verification work of these agents, leading to more differentiated verification agendas. Our starting hypotheses, therefore, are confirmed, although with nuances.

In this regard, we can identify an agenda in which COVID-19 can continue to occupy a leading role (Maldita's case) or a secondary role (Newtral's case) in the current stage (R.Q.1.1.). This result contrasts with other analyses obtained in different time periods, as in, for example, one year after the declaration of the state of alarm in Spain, when Newtral surpassed Maldita in verifications related to COVID-19 [29].

Therefore, COVID-19 has not disappeared from its leading role in the disinformation that continues to circulate today, although it has moved farther from the media spotlight. As a thematic subfield, hoaxes related to vaccines and their pernicious effects continue to be the main type of health disinformation detected linked to the topic of COVID-19 (R.Q.1.1.), coinciding with previous studies [29].

Nutrition (and related subtopics) has become a clear leader as one of the emerging disinformation hubs in the health field. The treatment of diseases and in particular disinformation about cancer is another of the topics that stand out (R.Q.1.2.). Previous and pre-pandemic studies also pointed to vaccines and certain diseases as significant thematic areas of disinformation owing to their presence on social media [38].

In this sense, we discuss the present and future social need for the main Spanish fact-checking platforms to either incorporate spaces specialized in health or to carry out greater fact-checking efforts in this area, given its proliferation and importance, as well as its consequences for society as a whole [39].

Throughout the study, there is a significant differentiation between the health fact-checking agenda of the generalist platforms (so-called throughout this study because they are dedicated to fact checking all types of topics, not only health) and that of the specialized platform (#SaludsinBulos). A greater number of hoaxes with a political perspective, i.e., relating the health issue in question to a secondary political interpretation, are found on generalist platforms than on specialized platforms. In the latter, health-related hoaxes are detected and combatted from a more informative perspective [40], and therefore, this disinformation refers directly or indirectly to behaviors, habits, or elements that affect the health of society (without any other dimension or interpretation).

This also applies to the geographic scope of reference and the impact of disinformation. In the case of the specialized platform, the hoaxes did not refer to or affect a specific and clear area, but in the generalist ones, there is an overwhelming tendency toward an international scope, where COVID-19 stands out as a thematic area, and secondary to disinformation with a national scope, showing a clear relationship to the current political agenda (with statements and gestures by relevant public figures in Spain).

This is also the case when it comes to the news sources used to fact-check fake news. On the generalist platforms, institutional sources stand out; on the specialized platform, scientific sources do.

In conclusion, different features of the health disinformation agenda emerge for generalists and specialized fact-checkers. Likewise, among generalist fact-checkers, the editorial

commitment to the role played by the analyzed health hoaxes and their typology, after the stage dominated by the pandemic, may be different (R.Q.1.2).

Most health hoaxes are manufactured, i.e., created from scratch, rather than being manipulated or reconfigured from real elements. In this sense, it is necessary to aim to improve public education regarding healthcare, such that health literacy itself will become the main defensive and critical tool available to citizens [41–44]. Both digital literacy and health literacy should be essential in Spain's public health policies for the coming years. Certain health and nutrition hoaxes, for example, recur over time.

In addition, the fact that the media are devoting more space and importance to health communication (a trend that seems to be slowly gaining traction over time) could also help to contain the impact of this type of disinformation. The existence of reliable online sites, especially on social media, would also help citizens identify safe resources, while other health information that circulates on the networks could be biased [40].

The image format continues to be predominant in this thematic area as well. However, it is also worth paying attention to the video format, which is a growing trend, as seen in a comparative look at previous studies [45].

Likewise, new social networks appear as platforms for the circulation of the detected hoaxes, such as TikTok or Telegram, while to date, WhatsApp has always been identified as the main platform for the circulation of hoaxes [5]. Taking into account the importance of new social media for young people [46], it would be advisable for scientific and authoritative sources to include them in their content dissemination strategies [47].

Regarding the drivers, although health hoaxes are not easily detectable, political or economic motivations may lie behind their circulation. Disinformation related to health can also be used to create a certain state of opinion, to stir up hate speech (relating, for example, certain foods and their countries of origin to negative elements), or to generate economic benefits. This study also indicates that the anti-vaccine movement related to COVID-19 continues to have an impact, which is consistent with the strategies of the anti-vaccine movements over time [26,48].

The present research is limited owing to its time frame and sociocultural context, as well as in terms of the fact-checking platforms selected. It is recommended, however, that this topic be continued in subsequent studies and research to elucidate the main aspects that characterize health disinformation and the composition of its agenda. Considering the importance of everything related to our health, determining how and what is being disseminated through the engineering of disinformation becomes of special relevance.

**Author Contributions:** Conceptualization: C.C.-S. and X.L.-G.; methodology, C.C.-S. and Á.V.; software, Á.V.; validation, C.C.-S. and X.L.-G.; formal analysis, C.C.-S. and Á.V.; investigation, C.C.-S.; resources, Á.V.; data curation, Á.V.; writing—original draft preparation, C.C.-S. and Á.V.; writing—review and editing, C.C.-S. and Á.V.; visualization, Á.V.; supervision, C.C.-S. and X.L.-G.; project administration, X.L.-G.; funding acquisition, X.L.-G. All authors have read and agreed to the published version of the manuscript.

**Funding:** This article is part of the R&D project "Digital Native Media in Spain: Strategies, Competencies, Social Involvement and (Re)Definition of Practices in Journalistic Production and Diffusion" (PID2021-122534OB-C21), funded by MCIN/10.13039/501100011033, and by the "ERDF a Way of Making Europe".

**Institutional Review Board Statement:** Not applicable.

**Informed Consent Statement:** Not applicable.

**Data Availability Statement:** The data presented in this study are available on request from the corresponding author. The data are not publicly available due to it is part of an ongoing research.

**Conflicts of Interest:** The authors declare no conflict of interest.

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
