# Peer review of "Fake News in the Post-COVID-19 Era? The Health Disinformation Agenda in Spain"

_societies, doi:10.3390/soc13110242_

Round 1
Reviewer 1 Report
Comments and Suggestions for Authors
The study is an interesting piece of work, with clear and appropriate methods. It explores an important topic, although it could be expanded as it does not examine a very large sample. I believe it should perhaps be complemented with a second phase in which some of the rumors are presented to better understand what it is about and how it affects, surpassing the quantitative study. It is a relevant piece of work, on the other hand. Perhaps the conclusions could be improved, and there is a lack of referencing or connection with the bibliography and other studies.
Author Response
Thank you very much for taking the time to review this manuscript. Please find the detailed responses below and the corresponding revisions/corrections highlighted/in track changes in the re-submitted files.
The study is an interesting piece of work, with clear and appropriate methods. It explores an important topic, although it could be expanded as it does not examine a very large sample. I believe it should perhaps be complemented with a second phase in which some of the rumors are presented to better understand what it is about and how it affects, surpassing the quantitative study. It is a relevant piece of work, on the other hand.
|
Thank you for your comments.
In the next study, we would like to add a new stage of exemplification of hoaxes and citizen perception. Thank you for the suggestion.
Note: all the changes made in the original manuscript are highlighted in yellow. |
Perhaps the conclusions could be improved, and there is a lack of referencing or connection with the bibliography and other studies.
|
The conclusions have been improved and the connection with other studies has been reinforced in Discussion and Conclusions section. |

Reviewer 2 Report
Comments and Suggestions for Authors
The article is interesting and useful in view of the special issue it deals with, and is also of interest for future work on misinformation and health. It certainly offers some limitations linked to the time frame, sample and context, hence the authors are encouraged to continue this line of study and broaden its scope in relation to this issue. It is also recommended to compensate for these shortcomings and enrich the work in other aspects, such as theoretical depth (state of the art, disinformation theory, etc.).
Therefore, the authors are encouraged to include a section on the state of the art that goes beyond the interesting background information provided in the introduction to situate the issue. Given the numerous theoretical contributions on the issue of misinformation linked to health and, specifically, to health issues during COVID (Brennen et al., 2020, García-Marin, 2020; Sánchez-Duarte, 2020; etc.), it is recommended that the authors carry out an exhaustive review of what these works have been, so that the reader can assess more precisely what the article's contribution is, in view of the conclusions offered.
Regarding methodology, It may be of interest to include some kind of hypothesis or starting premise to accompany the research objectives or questions. The sample analysed has been selected on the basis of a random time criterion, so this criterion should be better justified, taking into account that it generates a limited sample. On the other hand, the method described is clear and the article offers a wide range of results from a descriptive approach that serves to clarify and deepen health misinformation in the post-covid era.
The section devoted to conclusions and discussion is the most problematic due to two reasons: on the one hand, it offers results and devotes little space to their interpretation (already offered in the previous section), beyond linking these results to the research objectives; on the other hand, it would be nice to work more on the discussion to help the reader to know what the real contribution of the work is. All of this makes the article excessively descriptive and does not provide any reflections or deductions regarding the mechanisms for preventing misinformation in the specific field of health, or the mechanisms that promote it.
Some of the statements made in the conclusions section are too general or obvious and do not necessarily seem to derive from the analysis ("different pictures of the health disinformation agenda emerge for gen-335 eralist and specialised fact-checkers" , line 335-336; "Likewise, new social networks, such as TikTok or Telegram, are appearing as platforms for the circulation of the detected hoaxes, to which special attention should be paid considering that some of these hold significant importance for young people..." (lines 350-352).
Algunas afirmaciones deberían también ser mejor argumentadas a la luz de la teoría, caso de "This study also indicates that the anti-vaccine movement related to COVID-19 continues to have an impact and to stir up the networks with large amounts of disinformation aimed at creating distrust and fear about vaccines." (líneas 357-358).
Author Response
Thank you very much for taking the time to review this manuscript. Please find the detailed responses below and the corresponding revisions/corrections highlighted/in track changes in the re-submitted files.
The article is interesting and useful in view of the special issue it deals with, and is also of interest for future work on misinformation and health. It certainly offers some limitations linked to the time frame, sample and context, hence the authors are encouraged to continue this line of study and broaden its scope in relation to this issue. It is also recommended to compensate for these shortcomings and enrich the work in other aspects, such as theoretical depth (state of the art, disinformation theory, etc.).
|
Thank you for the suggestion. We have enriched the theoretical exposition and we have added specific references to disinformation theory and to the state of the art.
Note: all the changes made in the original manuscript are highlighted in yellow.
|
Therefore, the authors are encouraged to include a section on the state of the art that goes beyond the interesting background information provided in the introduction to situate the issue. Given the numerous theoretical contributions on the issue of misinformation linked to health and, specifically, to health issues during COVID (Brennen et al., 2020, García-Marin, 2020; Sánchez-Duarte, 2020; etc.), it is recommended that the authors carry out an exhaustive review of what these works have been, so that the reader can assess more precisely what the article's contribution is, in view of the conclusions offered.
|
We have added a specific revision about disinformation and pandemic in the Introduction section.
|
Regarding methodology, It may be of interest to include some kind of hypothesis or starting premise to accompany the research objectives or questions. The sample analysed has been selected on the basis of a random time criterion, so this criterion should be better justified, taking into account that it generates a limited sample. On the other hand, the method described is clear and the article offers a wide range of results from a descriptive approach that serves to clarify and deepen health misinformation in the post-covid era.
|
Regarding methodology, we have included the starting premise to accompany the research questions. The selection of the sample has been better explained.
|
The section devoted to conclusions and discussion is the most problematic due to two reasons: on the one hand, it offers results and devotes little space to their interpretation (already offered in the previous section), beyond linking these results to the research objectives; on the other hand, it would be nice to work more on the discussion to help the reader to know what the real contribution of the work is. All of this makes the article excessively descriptive and does not provide any reflections or deductions regarding the mechanisms for preventing misinformation in the specific field of health, or the mechanisms that promote it.
|
The real contribution of the work has been better explained. We have added at the beginning of the Discussion and Conclusions section a paragraph for explaining the singularities of the contribution of the work. We have added reflections about the mechanisms for preventing misinformation too. |
Some of the statements made in the conclusions section are too general or obvious and do not necessarily seem to derive from the analysis ("different pictures of the health disinformation agenda emerge for gen-335 eralist and specialised fact-checkers" , line 335-336; "Likewise, new social networks, such as TikTok or Telegram, are appearing as platforms for the circulation of the detected hoaxes, to which special attention should be paid considering that some of these hold significant importance for young people..." (lines 350-352).
|
That different things emerge from the treatment of health misinformation on generalist or specialized fact checking platforms is not obvious. It is the first time that it has been proved. Dealing with the same topic, the agenda of priority topics could be similar.
Regarding social media, the reflection has been nuanced to enrich it. |
Algunas afirmaciones deberían también ser mejor argumentadas a la luz de la teoría, caso de "This study also indicates that the anti-vaccine movement related to COVID-19 continues to have an impact and to stir up the networks with large amounts of disinformation aimed at creating distrust and fear about vaccines." (líneas 357-358).
|
This statement has been better argued. Thank you. |
